# Removal of Arsenate from Contaminated Water via Combined Addition of Magnesium-Based and Calcium-Based Adsorbents

**Hajime Sugita** *[ORCID], **Terumi Oguma, Junko Hara, Ming Zhang** [†] **and Yoshishige Kawabe**

Geological Survey of Japan, National Institute of Advanced Industrial Science and Technology (AIST), Central 7, 1-1-1 Higashi, Tsukuba 305-8567, Japan
* Correspondence: hajime.sugita@aist.go.jp
† Deceased author 16 July 2022.

**Abstract:** The effects of the combined addition of Mg- and Ca-based adsorbents (MgO, Mg(OH)$_2$, MgCO$_3$, CaO, Ca(OH)$_2$, and CaCO$_3$) were systematically tested for improving arsenic-removal performance and inhibiting the leaching of base material components from the adsorbent. Arsenic-removal tests were conducted with each single type or combination of two types of adsorbents. Results obtained after the combined-addition tests were compared with those obtained from the single-addition test with each adsorbent. The arsenic-removal performance improved in most combined additions but decreased in certain combined additions of MgO or Mg(OH)$_2$ with Ca-based adsorbents. The arsenic-removal performance of the combined addition of MgCO$_3$ and Ca(OH)$_2$ was the highest. The combination of Mg-based adsorbents with CaO or Ca(OH)$_2$ inhibited Mg-leaching, whereas that of CaO or Ca(OH)$_2$ with MgCO$_3$ inhibited Ca-leaching. Improvement in arsenic-removal performance for the combination of MgCO$_3$ with CaO or Ca(OH)$_2$ was caused by the incorporation and co-precipitation with arsenic when Mg(OH)$_2$ and CaCO$_3$ were produced. MgCO$_3$-Ca(OH)$_2$ and MgCO$_3$-CaO are recommended for both arsenic removal and environmental adsorbent stability that can be effectively applied over a wide range of arsenic concentrations.

**Keywords:** arsenic removal; adsorption; leaching; magnesium compounds; calcium compounds

## 1. Introduction

Arsenic (As) is considerably toxic to humans and sourced not only from industrial and mine wastewater but also from natural groundwater. The World Health Organization (WHO) guidelines on drinking water quality introduced a provisional As guideline value of 0.01 mg/L and important points on its properties and health effects [1]. Consuming As-containing drinking water and food can cause chronic arsenicism, leading to dermal lesions, peripheral neuropathy, skin cancer, bladder and lung cancers, and peripheral vascular disease, as well as acute As intoxication [1]. As-contaminated groundwater is often directly consumed, especially in developing regions such as Southern and Southeast Asia [2–16], Western and Southern Africa [17,18], and Latin America [19–23].

Arsenic speciation and the effects of As on human health are detailed in Singh et al. [24]. The chemical speciation of inorganic arsenic species with strong toxicity among them is important for health effects. Common inorganic arsenic species include arsenate As(V) and arsenite As(III). Inorganic arsenic species are biomethylated in the human body and converted to monomethylarsonic acid (MMA) and dimethylarcinic acid (DMA), which are less toxic than inorganic arsenic. The order of toxicity of arsenicals is as follows: MMA(III) > As(III) > As(V) > MMA(V) = DMA(V). Monomethylarsonic acid (MA(III)) is a highly toxic intermediate product created during arsenic biotransformation. Prolonged intake of drinking water containing inorganic As causes various adverse health effects such as skin lesions, cardiovascular disease, neurological effects, chronic lung disease,

cerebrovascular disease, reproductive disease, adverse renal affects, developmental abnormalities, hematological disorders, diabetes mellitus and cancers of skin, lung, liver, kidney and bladder.

The prevention of arsenic health hazards can only be achieved by properly treating As-contaminated water. There are many studies on As treatment methods such as membrane filtration, adsorption, coagulation, ion exchange, photocatalysis and photoelectrocatalysis. In developing countries, treatment methods involving inexpensive adsorbents are usually more likely to be used owing to economic and operational constraints. In the adsorption method, various adsorbents such as activated carbon, silica gel, iron-based adsorbents, and aluminum-based adsorbent are used. Some recent research on adsorbents for arsenic removal are presented below.

Fe-based adsorbents are popular materials that are often used for arsenic removal, and the arsenic adsorption mechanism has also been investigated. Liu et al. [25] investigated the adsorption mechanism of As(V) and As(III) on magnetite nanoparticles (MNPs). MNPs were synthesized using a modified Fe(II) and Fe(III) coprecipitation method under $N_2$ protection. The specific surface area of MNPs was 39 $m^2$/g. Although near-spherical primary particles with an average diameter of 34 nm, MNPs existed as larger aggregates under adsorption experimental condition. The aggregate size was 2.57 μm following a 30 min sonication, which increased to 5.1 μm after 24 h of shaking. As(V) and As(III) adsorption on MNPs reached equilibrium around 120 min. The adsorption equilibration time was set at 24 h to ensure complete reactions. Their experiments were performed at various initial As concentration, pH (5.0–9.0), ion strength (0–100 mM $NaNO_3$) and temperature (10 to 40 or 50 °C). Their isotherm data were fitted with the linearized Langmuir equation and the maximum adsorption capacity slightly increased with increasing temperature. As(V) adsorption decreased monotonically with increasing pH. As(III) adsorption also started to decrease at pH greater than 7.0. As(V) and As(III) adsorptions were hardly affected by ionic strength. No major redox reaction occurred for the As(V) or As(III) adsorbed on MNPs under anoxic experimental conditions. Conversely, dramatic redox reactions occurred upon exposure to air during the overnight drying process. They described that the possible role of reactive Fe(II) atoms in the redox transformation of adsorbed As should not be ignored, because Fe atoms in magnetite are in Fe(II) and Fe(III) mixed-valence states [25]. A new type of Fe-based adsorbent processed into an easy-to-use shape has also been proposed. Lou et al. [26] reported the removal of As(V) using sponges loaded with Fe-based adsorbents. They were synthesized a composite material cube-shaped open-celled cellulose sponge loaded superparamagnetic iron oxide nanoparticles (SPION). To assess the As(V) adsorption performance of the adsorbents, batch adsorption experiments with various initial As(V) concentrations (0–800 mg/L), contact times (up to over 1400 min), and different temperatures (293 and 343 K) were performed. The solution pH was adjusted to 3.6 prior to the adsorption experiments to optimize the adsorption. The adsorption tests were performed by adding the sponge-loaded SPION (0.2 g) to As(V) solution (25 mL). From the results of the adsorption experiments, the following facts were mainly clarified: (1) The adsorption capacity increased when increasing the As(V) initial concentration. (2) The adsorption at 293 K was better than at 343 K. (3) The adsorption capacity became almost constant after 60 min. (4) The adsorption capacity was 69.68 mg/g for the initial As(V) concentration of 800 mg/L. In addition, they described that the best model for their adsorption isotherm data was Freundlich, which highlights the importance of the heterogeneous surface of the adsorbents. Moreover, from the XANES spectrum, they determined that As(V) was not reduced to As(III) after being adsorbed. Adsorption–desorption cycle experiments (initial As(V) concentration was 200 mg/L) were also performed. Then, they reported that the cube adsorbent maintained high adsorption capacity even after 5 adsorption–desorption cycles [26]. Noteworthy among more recent works on Fe-based adsorbents has been on the adsorbent composed of activated carbon (AC) and $Fe_3O_4$ [27]. They made AC by carbonizing powdered sugarcane bagasse mixed with $H_3PO_4$. Then, the AC and a solution including $FeCl_3 \cdot 6H_2O$ were mixed and hydrothermally treated using

an autoclave to prepare a $Fe_3O_4/AC$ composite adsorbent. Their As adsorption was more accurately described by the Langmuir isotherm, as compared to the Freundlich isotherm. They concluded that the number of active sites on the composite surface was limited, and a monolayer of As(III) was formed over the homogenous composite surface [27].

Another interesting study on As removal by novel adsorbents [28] prepared novel gelatin-PVA/$La_2O_3$ (GPL) composite by copolymerization of polyvinyl alcohol (PVA) and gelatin in the presence of $La_2O_3$ using glutaraldehyde as a cross-linker. The GPL composite proved to be a superior adsorbent because it could effectively remove both As(V) and As(III) from real wastewater samples and because it was reusable [28]. In addition, studies on manganese oxide adsorbents modified with transition elements have also been reported. Zhang et al. [29] investigated a cobalt (Co)-doped hausmannite (one of Mn oxide minerals) for the removal of As(III) and As(V) from water. The Co-doped hausmannite was synthesized from manganese sulfate ($MnSO_4 \cdot H_2O$), cobalt sulfate ($CoSO_4 \cdot 7H_2O$), and sodium hydroxide (NaOH). Three Co-doped hausmannite samples with initial Co/Mn molar ratios of 0, 0.05 and 0.10 (HM, CoH5, and CoH10) were prepared using the co-precipitation method. The average particle size was 85, 131, and 103 nm for HM, CoH5, and CoH10. Correspondingly, the specific surface area was 12.4, 1.7 and 11.8 $m^2/g$, respectively. As(V) adsorption experiments, As(V) adsorption isotherm experiments, and As(III) oxidation experiments were performed. For As(V) adsorption experiments, the initial As(V) concentration was 7.5 mg/L and adsorbent concentration of 1 g/L, ionic strength (0.01 mol/L, 0.1 mol/L, and 0.2 mol/L), and reaction pH (4.5, 5.5, 6.5 and 7.5) at 25 °C for 24 h. Adsorption isotherm experiments were conducted with initial As(V) concentrations of 0–60 mg/L. For As(III) oxidation experiments, the initial As(III) concentration was of 15.8 mg/L. The results of these experiments have revealed the following: ionic strength had no effect on the adsorption of As(V) onto the mineral surface. The As(V) adsorption density decreased, as pH increased from 4.5 to 7.5. The As(V) removal by Co-doped samples from the solution gradually decreased with increasing Co-doping level. The Freundlich model significantly better fit the data than did the Langmuir model, suggesting that the active sites on the Co-doped samples were energetically heterogeneous. As(III) was oxidized to As(V) by Mn(III) and then adsorbed on the mineral surface. During the As(III) oxidation by these Co-doped samples under the experimental conditions, large amounts of Mn(II) and Co(II) were released. EXAFS analysis revealed that only As(V) was adsorbed on the mineral surface [29].

In addition to the above studies on Fe-based adsorbents and so on, many studies on Mg-based and Ca-based adsorbents have been conducted for the purpose of As removal. Park et al. [30] used magnesium chloride or magnesium sulfate to remove As(V) from a molybdenum oxide processing plant liquid containing approximately 70 g/L of Mo(VI) and 470 mg/L of As(V). The addition of MgO as a precipitating agent was also tested. They reported that As(V) could be removed to less than 5 mg/L at pH 10.2, resulting in a pure Mo(VI) liquid by adding magnesium at a Mg:As molar ratio of at least 2:1. Addition of either $MgCl_2$ and $MgSO_4$ resulted in precipitation of $Mg_3(AsO_4)_2$ and also $Mg(OH)_2$ at pH 9–11. This formed $Mg(OH)_2$ could also adsorb As(V). No such effect was observed with the addition of MgO. Park et al. also performed the As-removal tests using As(III) synthetic solution (450 mg/L as As). Adding 0.3 mol/L (24 g/L) of Mg, the pH range of 6–12 was tested. As(III) reached approximately 20 mg/L at pH 11. Removal of As(III) after oxidation by addition of $H_2O_2$ (a $H_2O_2$/As molar ratio of 3:1) was also tested. Within the range of pH 9–11, the residual As(V) after 20 min was less than 5 ppm for the Mg/As ratio of 3/1. [30]. Tresintsi et al. [31] suggested a procedure for the regeneration of iron oxyhydroxide arsenic adsorbents by granulated MgO. In their method, the arsenic desorbed from spent Fe-based adsorbents using NaOH aqueous solution was re-adsorbed on MgO. The optimum conditions of MgO application and the arsenic adsorption mechanism were examined through batch adsorption tests. A commercial fused MgO was used in the form of fine powder (<63 μm) for batch adsorption tests and granulated material (100–250 μm) for regeneration column tests. The surface

morphological characteristics for the used MgO were a specific surface area of 59 $m^2/g$, a pore volume 0.14 mL/g, and a mean pore diameter156 Å. The corresponding characteristics for the FeOOH were 155 $m^2/g$, 0.23 mL/g, and 30 Å. In batch adsorption experiments, around 20–25 mg of fine powder of MgO were dispersed in 200 mL of As(III) or As(V) solutions inside flasks and the solutions were stirred at 20 °C for 24 h. Initial As(III) and As(V) concentrations varied between 0.25 and 12.5 mg/L. The tests were performed at pH values 10–12. The optimum pH for As(V) adsorption was 10, where a maximum adsorption capacity 59.4 mg-As(V)/g was calculated for residual concentration near 5 mg/L. In the case of As(III), the removal capacity is maximized at pH 11, where around 50 mg As(III)/g could be adsorbed for residual concentration 3 mg/L. As K-edge EXAFS spectra indicated a high probability of adsorption of As(V) and As(III) on $Mg(OH)_2$ produced by hydrolysis of MgO [31]. Yu et al. [32] made a porous hierarchically micro/nanostructured MgO for As removal. Their MgO precursors were precipitates formed by mixing $Mg(NO_3)_2$ and $K_2CO_3$ solutions (at 293 K aging for 2 h). Two types of MgO precursors (flower-like and nest-like MgO precursors) were made by changing concentrations of $K_2CO_3$ (1 M and 0.5 M). The MgO precursors were calcined at 973 K for 4 h to produce two types of MgO. Using XRD identified the MgO precursors to be hydromagnesites. The two MgO precursors were assigned to flower-like hydromagnesite (F-hydromagnesite) and nest-like hydromagnesite (N-hydromagnesite), respectively. Correspondingly, the flower and nest-like MgO samples are assigned to F-MgO and N-MgO, respectively. The BET surface area of F- and N-hydromagnesite were approximately 21 and 18 $m^2/g$. Those of F- and N-MgO were approximately 33 and 25 $m^2/g$. For the comparisons of arsenic adsorption performance between MgO precursors and MgO, the initial As(III) and As(V) concentrations were approximately 4.6 and 7.2 mg/L, respectively. The adsorbent dose was 0.3 g/L in a comparison study. The adsorption capacity of two MgO was much higher than that of the hydromagnesites. The adsorption capacity of F-MgO and N-MgO for As(III) was approximately 252 mg/g and 644, respectively. That of F-MgO and N-MgO for As(V) was approximately 344 and 379 mg/g, respectively [32]. Opiso et al. [33] investigated the different mineral phases formed at alkaline condition in the Mg-Si-Al system and the sorption behavior of arsenate during and after mineral formation. In addition, the desorption of co-precipitated and adsorbed arsenate was conducted using phosphate-bearing solution. Their minerals were synthesized by mixing various volume ratios of Mg, Si, and Al solutions ($Na_2SiO_3$, $Mg(NO_3)_2 \cdot 6H_2O$ and $Al(NO_3)_3 \cdot 9H_2O$) at room temperature and 50 °C, respectively. The sorption of arsenate was investigated during and after mineral formation at alkaline conditions (around pH11). For co-precipitation experiments, arsenate solution were added instantaneously during the mixing of Mg, Si, and Al solutions to be 100 mg/L of As(V) concentration. In the case of adsorption experiments, the same amounts of As(V) were added after mineral formation. The suspension was then shaken for 7 days. The results revealed that brucite ($Mg(OH)_2$), hydrotalcite ($Mg_6Al_2(CO_3)(OH)_{16} \cdot 4(H_2O)$), and serpentine ($MgSi_2O_5(OH)_4$) have high uptake capacity for As(V) [33].

Camacho et al. [34] researched the effect of calcium addition as a stabilization agent on arsenic desorption from residues after ferric treatment of arsenic-contaminated water. They conducted laboratory and field tests using a calcium agent which was lime (CaO or $Ca(OH)_2$). The calcium addition was found to reduce arsenate leaching from ferric residuals prepared in their laboratory. The treatment residual field sample was a granular ferric hydroxide material used for arsenic removal from groundwater. Lime as calcium hydroxide was used as a binder for solidification/stabilization of arsenic in the field sample and arsenic stabilization was achieved with excess calcium addition (6 g per 10 g of air-dried treatment residual) [34]. Montes-Hernande et al. [35] investigated the removal of oxyanions such as arsenic from an aqueous solution using carbonation of $Ca(OH)_2$ under moderate pressure ($P_{CO2}$ = 20 bar) and temperature (30 °C). They placed one liter of high-purity water, 20 g of commercial portlandite $Ca(OH)_2$, 0 to 250 mg of sodium selenite pentahydrate $Na_2SeO_3 \cdot 5(H_2O)$, sodium selenate $Na_2SeO_4$, sodium acid arsenate heptahydrate $Na_2HAsO_4 \cdot 7(H_2O)$, and monosodium phosphate $NaH_2PO_4$ in a

titanium reactor. The solid particles were immediately dispersed by mechanical stirring (400 rpm) at 30 °C. Then, a 20 bar of $CO_2$ was injected in the reactor. At the end of the experiment, the reaction cell was rapidly depressurized for about 5 min and the autoclave was disassembled. Then, they reported that the $Ca(OH)_2$ carbonation reaction allowed for the successful removal of selenite (>90%), arsenate (>78%), and phosphate (almost 100%) from synthetic solutions [35]. Olyaie et al. [36] evaluated $CaO_2$ nanoparticles synthesized for removing As (III) from contaminated water. $CaO_2$ is one of the oxidants and decomposes in high humidity to produce $Ca(OH)_2$ and $H_2O_2$. The diameter of $CaO_2$ nanoparticle was 15–25 nm. The removal efficiency was enhanced by increasing the $CaO_2$ nanoparticles' dosage and reaction time. Up to 88% removal efficiency for arsenic was obtained by nanoparticles' dosage of 40 mg/L at time equal to 30 min and pH 7. However, decreased by increasing arsenic concentration and pH [36]. Hu et al. [37] investigated the effect of calcium on arsenate removal by electrocoagulation with aluminum electrode. The used calcium salt was $CaCl_2$. Their test conditions were an initial arsenic concentration of 10 mM (i.e., approximately 750 mg/L as As), an initial calcium addition concentration of 0–2 mol ratio to the initial arsenic concentration (i.e., 0–800 mg/L as Ca), and a reaction time of 40 min. The addition of calcium salt dramatically improved the removal efficiency of As(V). They concluded that this was due to calcium ions neutralizing the negative surface charge of the precipitates and increasing the As-O binding energy. In addition, they reported that the addition of calcium also prevented the formation of a deposit layer on anode surface which caused an increase of applied potential and a decrease in the concentration of dissolve Al [37].

As summarized above, Mg- and Ca-based adsorbents are expected to be so effective for As removal, as well as Fe-based adsorbents and so on. Furthermore, Mg and Ca compounds, which are the base materials of Mg- and Ca-based adsorbents, are abundantly available and are generally cheaper than Fe and Al. In addition, Mg and Ca components that may leach from Mg-based and Ca-based adsorbents are not harmful to humans or animals. However, the As-removal performance and/or the environmental stability of the adsorbents may deteriorate when the base materials leach from the system. Arsenic removal using individual Mg-based and Ca-based adsorbents has been studied by many researchers [30–37], but there has been little research on As removal using various combinations of Mg-based and Ca-based adsorbents. Furthermore, preliminary results with respect to the present study suggest the possibility of inhibiting the leaching of base materials by combining specific Mg- and Ca-based adsorbents. Investigating how As removal changes with the combination of different types of adsorbents will provide important guidance for designing novel treatments with high performance. This study aimed to find improved sustainable adsorbent combinations with both high As-removal performance and high environmental stability. The Mg- and Ca-based adsorbents were compared individually and in combination with each other in a full factorial experiment. Finally, improvements in the As-removal performance and environmental stability of the adsorbents using the combined-addition method were evaluated.

## 2. Materials and Methods

The reagents listed in this article were purchased from FUJIFILM Wako Pure Chemical Corporation (formerly Wako Pure Chemical Industries, Ltd., Osaka, Japan), unless specified otherwise.

### 2.1. Mg- and Ca-Based Adsorbents

In this study, to clarify the effects of arsenic-removal performance and base-materials leaching behavior due to the difference in the chemical compositions of the adsorbents, pure analytical grade powder reagents were used without processing, as adsorbents. The powder reagents of Mg and Ca oxides, hydroxides, and carbonates were used as Mg-based ($MgO$, $Mg(OH)_2$, and $MgCO_3$) and Ca-based ($CaO$, $Ca(OH)_2$, and $CaCO_3$) adsorbents. The measured Mg content $\alpha_{Mg}$ (%), the measured Ca contents $\alpha_{Ca}$ (%), the reagent purity

(obtained from $\alpha_{Mg}$ and $\alpha_{Ca}$) $P$ (%), the median particle size $D_{p50}$ (μm), and the Brunauer–Emmett–Teller (BET) surface area $S_{BET}$ (m$^2$/g) are shown in Table S1. The data on the table, except for $CaCO_3$, were obtained from a previous study [38].

The $\alpha_{Ca}$ value of $CaCO_3$ was determined by dissolving $CaCO_3$ in 1 M HCl and then measuring Ca using ICP-AES (SII SPS3500DD). The $D_{p50}$ value of $CaCO_3$ was determined using a particle size analyzer (Microtrac HRA 9320-X100, Nikkiso Co., Ltd., Tokyo, Japan). The $S_{BET}$ value of $CaCO_3$ was determined using a specific surface area measurement instrument (BELSOOP-miniX, MicrotracBEL Corp., Osaka, Japan).

### 2.2. Synthetic As(V)-Contaminated Water

Powdered disodium hydrogen arsenate heptahydrate ($Na_2HAsO_4 \cdot 7H_2O$, 99%) was dissolved in deionized water to create our As(V) stock solution (2000 mg-As/L). One part of the stock solution was diluted with deionized water to prepare a 1 or 10 mg-As/L solution. This solution was used as synthetic As(V)-contaminated water after adjusting the pH to near neutral using hydrochloric acid. The initial pH ($pH_0$) was taken as that of the synthetic As(V)-contaminated water immediately before adding the adsorbent. The initial As concentration ($C_{AS0}$ in mg/L) was taken as that in the synthetic As(V)-contaminated water immediately before adding the adsorbent, and was either 1 or 10 mg/L. The pH meter and electrode used in this study were LAQUA F-72 and a Micro ToupH Electrode 9618S-10D, respectively, manufactured by HORIBA, Ltd. (Kyoto, Japan). The As in the solution was quantified by ICP-MS (Agilent 7700X).

### 2.3. As-Removal Tests

As-removal tests were classified as either single-addition tests, which consisted of one type of adsorbent, or combined-addition tests, which consisted of two types of adsorbents.

One or two types of the adsorbents mentioned in Section 2.1 were weighted into a 50-mL polypropylene centrifuge tube. The amount of each adsorbent to be weighed was to become a set value of adsorbent addition concentration when 50 mL of the liquid was added. The centrifuge tube containing the adsorbent was sealed immediately after adding 50 mL of the above synthetic As(V)-contaminated water and shaken in a constant-temperature shaker (approximately 180 rpm at 20–25 °C). The tube was centrifuged after shaking for 24 h. The supernatant was filtered through a syringe filter with a pore diameter of 0.45 μm and collected in a polypropylene vessel. The pH of the filtrate (treated water) was immediately measured and taken as the final pH ($pH_f$). As, Mg, and Ca in the treated water were quantified using the ICP-MS and the ICP-AES, respectively, and were denoted as $C_{AS}$, $C_{Mg}$, and $C_{Ca}$, respectively.

The adsorbent addition concentration in the As-removal tests was set on a mass basis. $W_{Ad}$ was the amount of adsorbent added to the synthetic As(V)-contaminated water in g, and $V$ was the liquid volume (in L) of the synthetic As-contaminated water.

In the single-addition tests of one type of adsorbent, the adsorbent addition concentration, $W_{Ad}/V$ (g/L), was set to 0.2 and 0.4 g/L.

In the combined-addition tests of two types of adsorbents, the mass-based total adsorbent addition concentration, $\Sigma W_{Ad}/V$ (g/L), was set to 0.4 g/L (each 0.2 g/L).

### 2.4. Definitions of Mg- and Ca-Residual Ratios

As some of the constituents of the adsorbent can leach into the liquid, the amount of adsorbent remaining as a solid is reduced from the initial amount. As the Mg and Ca components are leached out from the Mg-based and Ca-based adsorbents, the amounts of Mg and Ca remaining as a solid in the liquid are calculated as the Mg-residual ratio $R_{Mg}$ [%] and $R_{Ca}$ [%], using the Equations (1) and (2).

$$R_{Mg} = 100 - (C_{Mg} \times 1000)/[W_{Ad}/V \times (\alpha_{Mg}/100)] \times 100. \tag{1}$$

$$R_{Ca} = 100 - (C_{Ca} \times 1000)/[W_{Ad}/V \times (\alpha_{Ca}/100)] \times 100. \tag{2}$$

## 3. Results

### 3.1. Single Addition of One Type of Adsorbent

Table S2 shows the results obtained from the single-addition tests with one type of adsorbent at $C_{AS0}$ = 1 mg/L. To confirm the reproducibility of the experimental data, test nos. 1–6 were conducted three times each, the results of which are presented in Table S2. The table shows average values obtained from each test conducted thrice. In addition, Table S3 shows the relative standard errors $\varepsilon$ [%] for all six tests in Table S2. The standard errors of both $pH_0$ and $C_{AS0}$ for each test conducted thrice are zero, because the synthetic As(V)-contaminated water used in the tests were the same. The values of $\varepsilon$ for $W_{Ad}/V$, were within 1.2%, which can be attributed to the accuracy of weighing the adsorbent. The values of $\varepsilon$ for $pH_0$, $C_{AS}$, $C_{Mg}$, and $C_{Ca}$ were within 0.1%, 5%, 4%, and 3%, respectively. These results indicate that the reproducibility of the experimental data was good.

As shown in Table S2, regardless of $W_{Ad}/V$, the magnitude of the relationship of $pH_f$ was $CaCO_3$ < $Mg(OH)_2$ < $MgCO_3$ < $MgO$ < $Ca(OH)_2$ = $CaO$. There was no clear difference in $pH_f$ for adsorbents other than CaO and $Ca(OH)_2$ depending on $W_{Ad}/V$, but the $pH_f$ for CaO and $Ca(OH)_2$ clearly increased with increasing $W_{Ad}/V$. At both 0.2 and 0.4 g/L of $W_{Ad}/V$, the Mg-based adsorbents exhibited an improved As-removal performance compared with the Ca-based adsorbents. However, the magnitude of the relationship of the As-removal performance of the adsorbent differed only slightly depending on $W_{Ad}/V$. The magnitude of the relationship of $C_{AS}$ at $W_{Ad}/V$ = 0.2 g/L was $Mg(OH)_2$ < $MgO$ < $MgCO_3$ < $Ca(OH)_2$ < $CaO$ < $CaCO_3$. The magnitude relationship of $C_{AS}$ at $W_{Ad}/V$ = 0.4 g/L was $MgO$ < $Mg(OH)_2$ < $MgCO_3$ < $CaO$ < $Ca(OH)_2$ < $CaCO_3$. Regardless of $W_{Ad}/V$, the magnitude of the relationships of $C_{Mg}$ and $C_{Ca}$ were $Mg(OH)_2$ < $MgO$ < $MgCO_3$ and $CaCO_3$ < $Ca(OH)_2$ < $CaO$, respectively.

Table S4 shows the results obtained from the single-addition tests with one type of adsorbent at $C_{AS0}$ = 10 mg/L. The tendencies of $pH_f$, $C_{Mg}$, and $C_{Ca}$ at $C_{AS0}$ = 10 mg/L were similar to those at $C_{AS0}$ = 1 mg/L, except for the following two points: (1) the $pH_f$ for $CaCO_3$, as with other Ca-based adsorbents, clearly increased with increasing $W_{Ad}/V$; (2) $C_{Ca}$ was slightly higher for $Ca(OH)_2$ than for CaO at $W_{Ad}/V$ = 0.4 g/L. In contrast, the tendency of $C_{AS}$ was significantly different from that at $C_{AS0}$ = 1 mg/L, except for the highest $C_{AS}$ for $CaCO_3$. At $W_{Ad}/V$ = 0.2 g/L, the magnitude of the relationship of $C_{AS}$ was $CaO$ < $MgO$ < $Mg(OH)_2$ < $MgCO_3$ < $Ca(OH)_2$ < $CaCO_3$. At $W_{Ad}/V$ = 0.4 g/L, the magnitude of the relationship of $C_{AS}$ was $Ca(OH)_2$ < $CaO$ < $MgO$ < $Mg(OH)_2$ < $MgCO_3$ < $CaCO_3$.

### 3.2. Combined Addition of Two Types of Adsorbents

Tables S5 and S6 show the results obtained from the combined-addition tests of two types of adsorbents at $C_{AS0}$ = 1 and 10 mg/L, respectively.

Regardless of $C_{AS0}$, the $pH_f$ was higher in the combined-addition tests between the Ca-based adsorbents than between the Mg-based adsorbents. In the combined-addition tests of one Mg-based with one Ca-based adsorbent, $pH_f$ tended to become relatively high with CaO or $Ca(OH)_2$, but relatively low with $CaCO_3$.

The $C_{Mg}$ in the combined addition-tests of two different types of the Mg-based adsorbents did not sum to the total of the $C_{Mg}$ in their single-addition tests and appeared to be closer to the higher values between the single tests. The $C_{Mg}$ tended to be lower in the combined additions of the Mg-based adsorbents with CaO or $Ca(OH)_2$ than in the single additions of each Mg-based adsorbent. The $C_{Mg}$ in the combined additions of the Mg-based adsorbents with $CaCO_3$ were not significantly different from those in the single additions of each Mg-based adsorbent.

The $C_{Ca}$ in the combined addition-tests of two different types of the Ca-based adsorbents appeared to be close to the sum of the $C_{Ca}$ in their single-addition tests and appeared somewhat higher in CaO–$CaCO_3$. There was no significant difference in $C_{Ca}$ between the combined addition of CaO or $Ca(OH)_2$ with MgO or $Mg(OH)_2$ and the single additions of each Ca-based adsorbent. In contrast, the $C_{Ca}$ in the combined additions of the Ca-based adsorbents with $MgCO_3$ tended to be lower than that in the single additions of one type of

each Ca-based adsorbent. In addition, the $C_{Ca}$ in the combined additions of $CaCO_3$ and the Mg-based adsorbents were lower than those in the single addition of $CaCO_3$.

Moreover, in contrast, the tendency of the $C_{AS}$ differed depending on $C_{AS0}$. At $C_{AS0} = 1$ mg/L, the $C_{AS}$ was significantly lower in the combined-addition tests of MgO-Mg(OH)$_2$, MgCO$_3$-Ca(OH)$_2$, MgCO$_3$-CaO, and Mg(OH)$_2$-CaO. At $C_{AS0} = 10$ mg/L, the $C_{AS}$ was significantly lower in the combined-addition tests of MgCO$_3$-Ca(OH)$_2$, MgCO$_3$-CaO, Mg(OH)$_2$-Ca(OH)$_2$, and CaO-Ca(OH)$_2$.

## 4. Discussion

The effects of improving As-removal performance and those of inhibiting Mg- and Ca-leaching via combined addition have been discussed below based on the data obtained from the As-removal tests conducted in this study.

### 4.1. As-Removal Performance

4.1.1. As-Removal Ratio

The As-removal ratio, $R_{AS}$ [%], was calculated as:

$$R_{AS} = (C_{AS0} - C_{AS})/C_{AS0} \times 100. \tag{3}$$

$R_{AS}$ values are summarized in Tables 1–3. Table 1 shows $R_{AS}$ for the single-addition As-removal tests.

**Table 1.** As removal ratio ($R_{AS}$) for single-addition As-removal tests.

| $W_{Ad}/V$ [g/L] | $C_{AS0}$ [mg/L] | MgO | Mg(OH)$_2$ | MgCO$_3$ | CaO | Ca(OH)$_2$ | CaCO$_3$ |
|---|---|---|---|---|---|---|---|
| 0.2 | 1 | 44.1 | 86.9 | 33.9 | 4.4 | 10.1 | 1.7 |
| 0.4 | 1 | 98.1 | 97.5 | 53.6 | 15.7 | 19.9 | 2.1 |
| 0.2 | 10 | 17.2 | 13.2 | 8.7 | 23.5 | 4.9 | 0.0 |
| 0.4 | 10 | 31.3 | 26.2 | 18.8 | 51.7 | 60.6 | 0.0 |

unit [%].

**Table 2.** As removal ratio ($R_{AS}$) for combined-addition As-removal tests at $C_{AS0} = 1$ mg/L.

| $\Sigma W_{Ad}/V = 0.4$ g/L | MgO | Mg(OH)$_2$ | MgCO$_3$ | CaO | Ca(OH)$_2$ | CaCO$_3$ |
|---|---|---|---|---|---|---|
| MgO | 98.1 | 99.9 | 98.8 | 85.5 | 57.5 | 65.3 |
| Mg(OH)$_2$ | 99.9 | 97.5 | 88.4 | 99.1 | 90.0 | 79.2 |
| MgCO$_3$ | 98.8 | 88.4 | 53.6 | 99.7 | 99.9 | 33.3 |
| CaO | 85.5 | 99.1 | 99.7 | 15.7 | 32.6 | 1.6 |
| Ca(OH)$_2$ | 57.5 | 90.0 | 99.9 | 32.6 | 19.9 | 13.6 |
| CaCO$_3$ | 65.3 | 79.2 | 33.3 | 1.6 | 13.6 | 2.1 |

unit [%].

**Table 3.** As removal ratio ($R_{AS}$) for combined-addition As-removal tests at $C_{AS0} = 10$ mg/L.

| $\Sigma W_{Ad}/V = 0.4$ g/L | MgO | Mg(OH)$_2$ | MgCO$_3$ | CaO | Ca(OH)$_2$ | CaCO$_3$ |
|---|---|---|---|---|---|---|
| MgO | 31.3 | 43.8 | 33.6 | 16.0 | 13.1 | 10.4 |
| Mg(OH)$_2$ | 43.8 | 26.2 | 21.0 | 38.0 | 58.1 | 12.2 |
| MgCO$_3$ | 33.6 | 21.0 | 18.8 | 63.9 | 83.7 | 9.8 |
| CaO | 16.0 | 38.0 | 63.9 | 51.7 | 60.1 | 42.5 |
| Ca(OH)$_2$ | 13.1 | 58.1 | 83.7 | 60.1 | 60.6 | 8.2 |
| CaCO$_3$ | 10.4 | 12.2 | 9.8 | 42.5 | 8.2 | 0.0 |

unit [%].

From Table 1, it is clear that $R_{AS}$ did not necessarily double when the adsorbent addition concentrations were doubled. Although CaCO$_3$ could not remove As even if the

adsorbent addition concentration was increased, the $R_{AS}$ of the other adsorbents increased as the adsorbent addition concentration increased. Some adsorbents, such as CaO, more than doubled their $R_{AS}$ when the adsorbent concentrations were only doubled. Overall, the $R_{AS}$ was higher for Mg-based adsorbents at $C_{AS0}$ = 1 mg/L and higher for Ca-based adsorbents (excluding CaCO$_3$) at $C_{AS0}$ = 10 mg/L. In general, the surface area of adsorbent can be an important parameter related to its adsorption capacity. The magnitude of $S_{BET}$ is CaCO$_3$ < CaO < MgO < Ca(OH)$_2$ < Mg(OH)$_2$ < MgCO$_3$, as shown in Table S1. Hence, the magnitude relationship of $R_{AS}$ among the adsorbents did not correspond to that of $S_{BET}$. Section 1 introduced previous studies on Mg-based and Ca-based adsorbents [30–37], but since the experimental conditions differed among their experiments, it is difficult to directly determine the order of superiority among their adsorbents. However, Yu et al. [32] compared two MgO and two hydromagnesites (i.e., MgCO$_3$) under the same experimental conditions. They reported that the As adsorption capacity of MgO was much higher than that of MgCO$_3$, but there was no significant difference in specific surface area among them. This superiority in As adsorption between MgO and MgCO$_3$ is consistent with our experimental results. Therefore, it is believed that the arsenic-removal performance is more strongly related to the chemical composition of the adsorbent than to the surface area.

Tables 2 and 3 show $R_{AS}$ for the combined addition of two types of adsorbents obtained from As-removal tests with the mass-based adsorbent addition concentration at $C_{AS0}$ = 1 and 10 mg/L, respectively. For convenience, the results of single additions at $W_{Ad}/V$ = 0.4 g/L are also included in Tables 2 and 3. In addition, each $R_{AS}$ of the combined additions is shown in Figure 1, based on the data in Tables 2 and 3. Figure 1a,b correspond to $C_{AS0}$ = 1 and 10 mg/L, respectively.

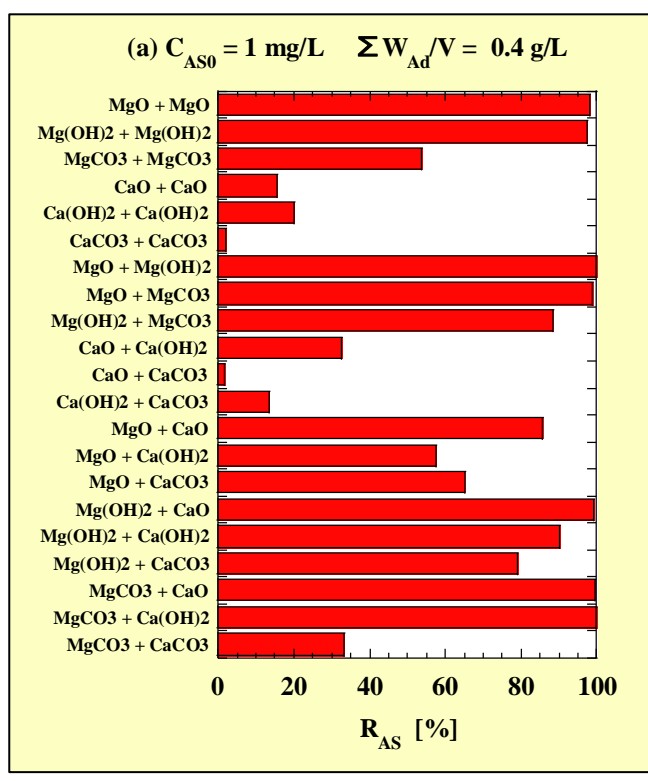
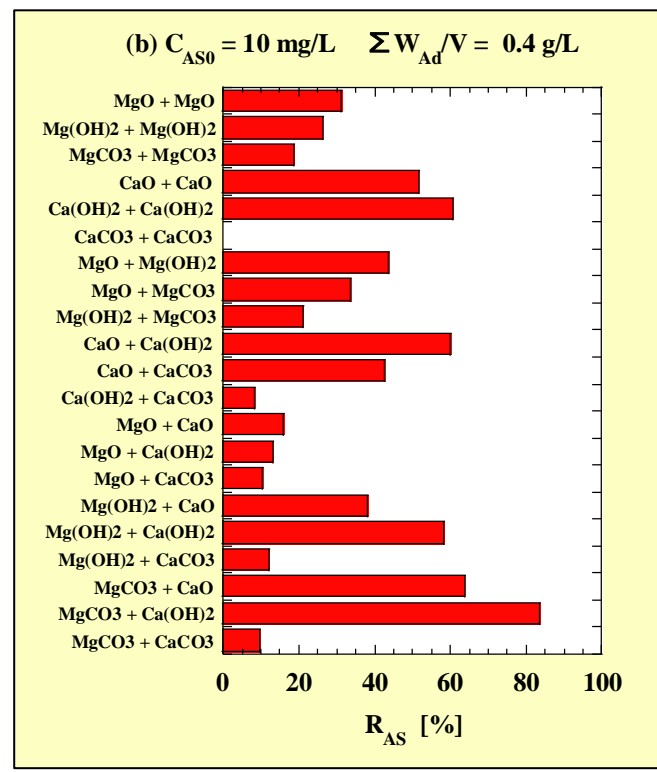

**Figure 1.** $R_{AS}$ for combined-addition at (**a**) $C_{AS0}$ = 1 mg/L and (**b**) $C_{AS0}$ = 10 mg/L. $R_{AS}$ = $(C_{AS0} - C_{AS})/C_{AS0} \times 100$.

At $C_{AS0}$ = 1 mg/L, the $R_{AS}$ of the combined additions of MgO-Mg(OH)$_2$, MgCO$_3$-Ca(OH)$_2$, MgCO$_3$-CaO, MgO-MgCO$_3$, Mg(OH)$_2$-CaO, MgO-MgO, Mg(OH)$_2$-Mg(OH)$_2$ and Mg(OH)$_2$-Ca(OH)$_2$ were high (>90%). At $C_{AS0}$ = 10 mg/L, the $R_{AS}$ of the combined

addition of $MgCO_3$-$Ca(OH)_2$ was the highest (>80%) and followed by those of $MgCO_3$-CaO, $Ca(OH)_2$-$Ca(OH)_2$, and CaO-$Ca(OH)_2$ (>60%).

From the above, the As-removal performance of the combined addition of $MgCO_3$-$Ca(OH)_2$ was the highest at both 1 and 10 mg/L of $C_{AS0}$.

### 4.1.2. Effects of Combined Addition on As-Removal Performance

When multiple adsorbents are added together, the $R_{AS}$ was typically not equal to the sum of the individual adsorbents $R_{AS}$ values. This apparent combined-adsorbent synergistic effect was evaluated using the following method. The higher of the two single-addition $R_{AS}$ values for each adsorbent of a combined test was subtracted from the $R_{AS}$ in the combined-addition test. The obtained difference was denoted as $\Delta R_{AS}$.

The $\Delta R_{AS}$ values are shown in Table 4 for $C_{AS0}$ = 1 mg/L and Table 5 for $C_{AS0}$ = 10 mg/L. Similar to Tables 2 and 3, the results from the single-additions, $W_{Ad}/V$ = 0.4 g/L tests are included. Similar to Figure 1, each $\Delta R_{AS}$ of the combined additions is shown in Figure 2, based on the data in Tables 4 and 5. Figure 2a,b correspond to $C_{AS0}$ = 1 and 10 mg/L, respectively.

**Table 4.** The combined-addition $R_{AS}$ minus the higher of the two single-addition $R_{AS}$ ($\Delta R_{AS}$) for As-removal tests at $C_{AS0}$ = 1 mg/L.

| $\Sigma W_{Ad}/V$ = 0.4 g/L | MgO | $Mg(OH)_2$ | $MgCO_3$ | CaO | $Ca(OH)_2$ | $CaCO_3$ |
|---|---|---|---|---|---|---|
| MgO | 54.0 | 13.1 | 54.7 | 41.4 | 13.4 | 21.2 |
| $Mg(OH)_2$ | 13.1 | 10.6 | 1.5 | 12.2 | 3.2 | −7.7 |
| $MgCO_3$ | 54.7 | 1.5 | 19.6 | 65.7 | 66.0 | −0.7 |
| CaO | 41.4 | 12.2 | 65.7 | 11.3 | 22.5 | −2.8 |
| $Ca(OH)_2$ | 13.4 | 3.2 | 66.0 | 22.5 | 9.8 | 3.5 |
| $CaCO_3$ | 21.2 | −7.7 | −0.7 | −2.8 | 3.5 | 0.4 |

unit [%].

**Table 5.** The combined-addition $R_{AS}$ minus the higher of the two single-addition $R_{AS}$ ($\Delta R_{AS}$) for As-removal tests at $C_{AS0}$ = 10 mg/L.

| $\Sigma W_{Ad}/V$ = 0.4 g/L | MgO | $Mg(OH)_2$ | $MgCO_3$ | CaO | $Ca(OH)_2$ | $CaCO_3$ |
|---|---|---|---|---|---|---|
| MgO | 14.1 | 26.6 | 16.5 | −7.5 | −4.0 | −6.7 |
| $Mg(OH)_2$ | 26.6 | 13.0 | 7.8 | 14.5 | 45.0 | −1.0 |
| $MgCO_3$ | 16.5 | 7.8 | 10.1 | 40.4 | 75.0 | 1.1 |
| CaO | −7.5 | 14.5 | 40.4 | 28.2 | 36.6 | 19.0 |
| $Ca(OH)_2$ | −4.0 | 45.0 | 75.0 | 36.6 | 55.8 | 3.3 |
| $CaCO_3$ | −6.7 | −1.0 | 1.1 | 19.0 | 3.3 | 0.0 |

unit [%].

At $C_{AS0}$ = 1 mg/L, the $\Delta R_{AS}$ of the combined additions of $MgCO_3$-$Ca(OH)_2$ and $MgCO_3$-CaO were the highest (>60%). At $C_{AS0}$ = 10 mg/L, the $\Delta R_{AS}$ of the combined addition of $MgCO_3$-$Ca(OH)_2$ was the highest (>70%). On the other hand, the $\Delta R_{AS}$ of the combined addition of $Mg(OH)_2$-$CaCO_3$ at $C_{AS0}$ = 1 mg/L and those of MgO-CaO and MgO-$CaCO_3$ at $C_{AS0}$ = 10 mg/L were significantly negative (<−5%), indicating that the As-removal performance of the combined addition was lower than that of each single addition.

From the above, it was found that many combined additions, especially $MgCO_3$-CaO and $MgCO_3$-$Ca(OH)_2$, improved As-removal performance; however, some combined additions of the Ca-based adsorbents, MgO and -$Mg(OH)_2$, decreased the As-removal performance. In the former two combined-additions ($MgCO_3$-CaO and $MgCO_3$-$Ca(OH)_2$), it is considered that $MgCO_3$ released $Mg^{2+}$ and $CO_3^{2-}$, and CaO or $Ca(OH)_2$ released $Ca^{2+}$ and $OH^-$, followed by reformations of $Mg(OH)_2$ and $CaCO_3$. Hence, it is suggested that the As-removal performance improved because the As components were taken in during the formations of $Mg(OH)_2$ and $CaCO_3$ (i.e., co-precipitation phenomenon).

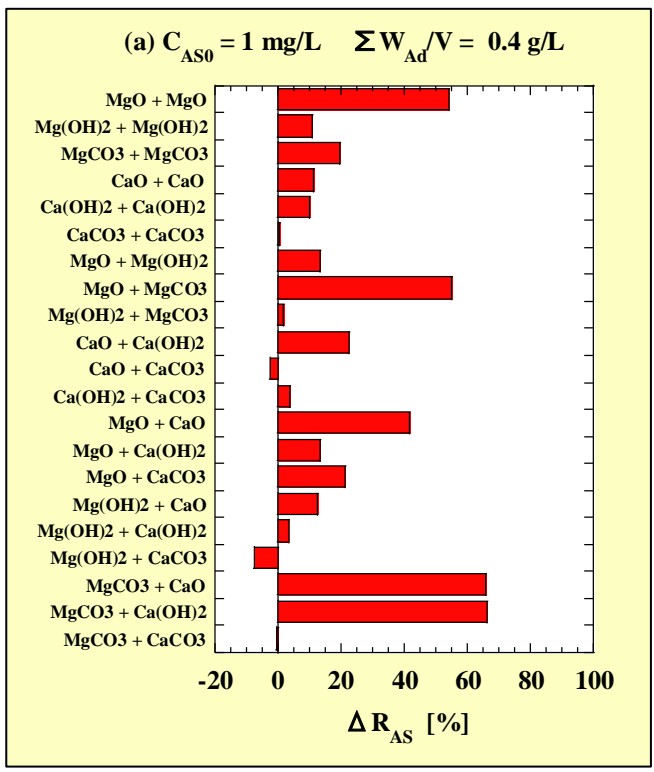
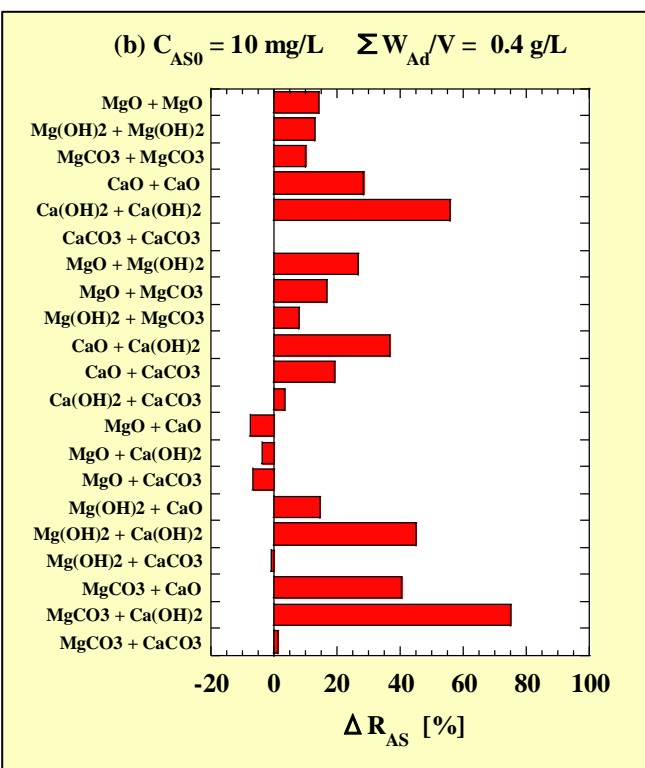

**Figure 2.** $\Delta R_{AS}$ for combined-addition at (**a**) $C_{AS0}$ = 1 mg/L and (**b**) $C_{AS0}$ = 10 mg/L. $\Delta R_{AS}$ = (the combined-addition $R_{AS}$) − (the higher of the two single-addition $R_{AS}$).

*4.2. Mg-Leaching Behavior*

4.2.1. Mg-Residual Ratio

$R_{Mg}$ obtained from the As-removal tests in this study are shown in Tables 6–8. However, it should be noted that the values of $R_{Mg}$ obtained from these calculations include the resolidified portion of Mg once leached out.

**Table 6.** Mg residual ratio ($R_{Mg}$) for single-addition As-removal tests.

| $W_{Ad}/V$ [g/L] | $C_{AS0}$ [mg/L] | MgO | Mg(OH)$_2$ | MgCO$_3$ | CaO | Ca(OH)$_2$ | CaCO$_3$ |
|---|---|---|---|---|---|---|---|
| 0.2 | 1 | 90.9 | 95.0 | 56.9 | - | - | - |
| 0.4 | 1 | 96.6 | 97.1 | 75.3 | - | - | - |
| 0.2 | 10 | 91.3 | 94.7 | 59.8 | - | - | - |
| 0.4 | 10 | 95.8 | 97.3 | 78.3 | - | - | - |

unit [%].

**Table 7.** Mg residual ratio ($R_{Mg}$) for combined-addition As-removal tests at $C_{AS0}$ = 1 mg/L.

| $\Sigma W_{Ad}/V$ = 0.4 g/L | MgO | Mg(OH)$_2$ | MgCO$_3$ | CaO | Ca(OH)$_2$ | CaCO$_3$ |
|---|---|---|---|---|---|---|
| MgO | 96.6 | 96.0 | 87.0 | 99.5 | 99.7 | 90.8 |
| Mg(OH)$_2$ | 96.0 | 97.1 | 82.7 | 100 | 100 | 95.0 |
| MgCO$_3$ | 87.0 | 82.7 | 75.3 | 99.7 | 98.6 | 58.5 |

unit [%].

Table 6 shows $R_{Mg}$ for the single addition of one type of adsorbent obtained from As-removal tests.

For all the Mg-based adsorbents, the $R_{Mg}$ of the higher adsorbent addition concentration were higher than that of the lower adsorbent addition concentration. Both the $R_{Mg}$ of MgO and Mg(OH)$_2$ were very high (>90%) in the range of the experimental conditions.

Tables 7 and 8 show $R_{Mg}$ for the combined addition of two types of adsorbents obtained from As removal tests at $C_{AS0}$ = 1 and 10 mg/L, respectively. For convenience, the results in the single additions of $W_{Ad}/V$ = 0.4 g/L have also been included in Tables 7 and 9. In addition, each $R_{Mg}$ of the combined additions is shown in Figure 3, based on the data in Tables 7 and 8. Figure 3a,b correspond to $C_{AS0}$ = 1 and 10 mg/L, respectively.

**Table 8.** Mg residual ratio ($R_{Mg}$) for combined-addition As-removal tests at $C_{AS0}$ = 10 mg/L.

| $\Sigma W_{Ad}/V$ = 0.4 g/L | MgO | Mg(OH)$_2$ | MgCO$_3$ | CaO | Ca(OH)$_2$ | CaCO$_3$ |
|---|---|---|---|---|---|---|
| MgO | 95.8 | 94.7 | 89.4 | 99.4 | 99.5 | 91.8 |
| Mg(OH)$_2$ | 94.7 | 97.3 | 85.2 | 100 | 100 | 95.2 |
| MgCO$_3$ | 89.4 | 85.2 | 78.3 | 97.4 | 95.3 | 64.6 |

unit [%].

**Table 9.** The combined-addition $C_{Mg}$ minus the higher of the two single-addition $C_{Mg}$ ($\Delta C_{Mg}$) for As-removal tests at $C_{AS0}$ = 1 mg/L.

| $\Sigma W_{Ad}/V$ = 0.4 g/L | MgO | Mg(OH)$_2$ | MgCO$_3$ | CaO | Ca(OH)$_2$ | CaCO$_3$ |
|---|---|---|---|---|---|---|
| MgO | −3.0 | −2.6 | 1.0 | −10.6 | −10.8 | 0.6 |
| Mg(OH)$_2$ | −2.6 | 0.5 | 0.7 | −4.2 | −4.2 | 0.0 |
| MgCO$_3$ | 1.0 | 0.7 | 2.5 | −22.3 | −21.8 | −0.2 |

unit [mg/L].

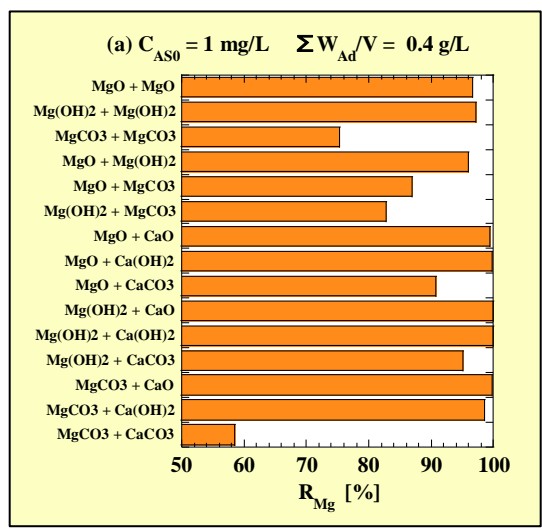
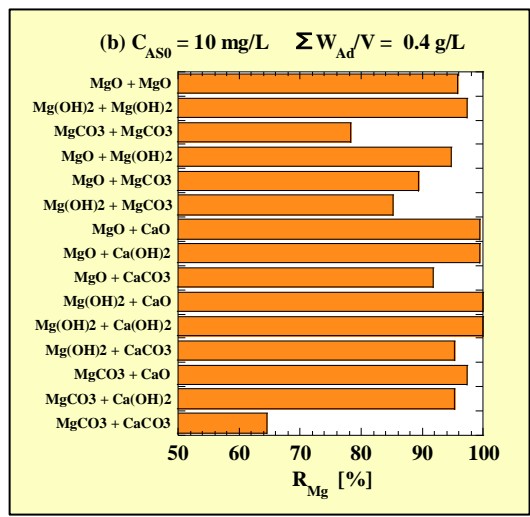

**Figure 3.** $R_{Mg}$ for combined-addition at (**a**) $C_{AS0}$ = 1 mg/L and (**b**) $C_{AS0}$ = 10 mg/L. $R_{Mg}$ = 100 − ($C_{Mg}$ × 1000)/($W_{Ad}/V$ × ($\alpha_{Mg}$/100)) × 100.

There was no difference in the tendency of $R_{Mg}$ depending on $C_{AS0}$. Most of the combined additions had an $R_{Mg}$ of >90%. Particularly, the $R_{Mg}$ of Mg(OH)$_2$-CaO or -Ca(OH)$_2$, and MgO-CaO or -Ca(OH)$_2$ were almost 100%. On the other hand, the $R_{Mg}$ of MgCO$_3$-CaCO$_3$ was the lowest (58.5% at $C_{AS0}$ = 1 mg/L and 64.6% at $C_{AS0}$ = 10 mg/L).

4.2.2. Effects of Combined Addition on Mg-Leaching Behavior

Even if the Mg-residual ratio was the same, the Mg leached amount differed depending on the type of Mg-based adsorbent. Therefore, to evaluate the effects of the combined addition on the Mg-leaching behavior, it was more appropriate to compare the amount of change in $C_{Mg}$. Then, the higher $C_{Mg}$ in the single-addition test ($W_{Ad}/V$ = 0.2 g/L) of the two types of adsorbents used in the combined-addition test ($\Sigma W_{Ad}/V$ = 0.4 g/L) was subtracted from the $C_{Mg}$ in the combined-addition test. The obtained difference was denoted as $\Delta C_{Mg}$. The lower the value of $\Delta C_{Mg}$, the higher the effects of inhibiting Mg-leaching.

The $\Delta C_{Mg}$ is shown in Table 9 for $C_{AS0}$ = 1 mg/L and Table 10 for $C_{AS0}$ = 10 mg/L, respectively. Similar to Figure 2, each $\Delta C_{Mg}$ of the combined additions is shown in Figure 4, based on the data in Tables 9 and 10. Figure 4a,b correspond to $C_{AS0}$ = 1 and 10 mg/L, respectively.

**Table 10.** The combined-addition $C_{Mg}$ minus the higher of the two single-addition $C_{Mg}$ ($\Delta C_{Mg}$) for As-removal tests at $C_{AS0}$ = 10 mg/L.

| $\Sigma W_{Ad}/V$ = 0.4 g/L | MgO | Mg(OH)$_2$ | MgCO$_3$ | CaO | Ca(OH)$_2$ | CaCO$_3$ |
|---|---|---|---|---|---|---|
| MgO | −0.2 | 0.4 | −2.3 | −9.8 | −9.8 | −0.4 |
| Mg(OH)$_2$ | 0.4 | 0.1 | −1.4 | −4.4 | −4.4 | −0.3 |
| MgCO$_3$ | −2.3 | −1.4 | 1.0 | −19.6 | −18.5 | −2.1 |

unit [mg/L].

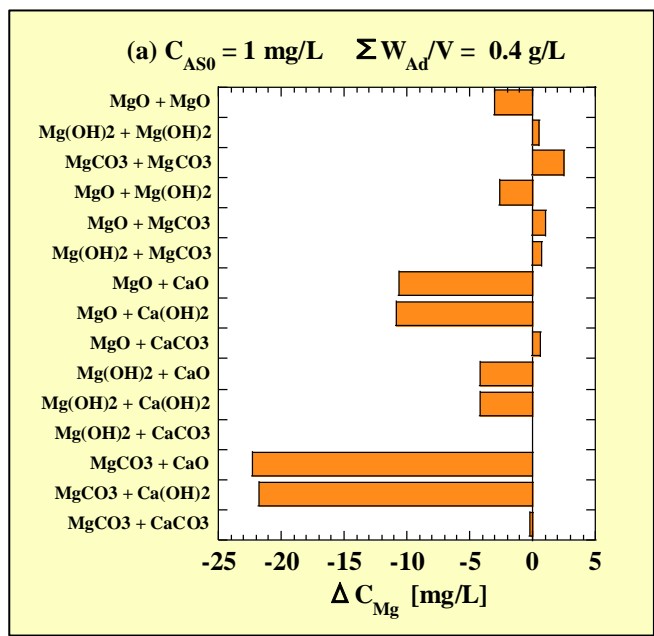 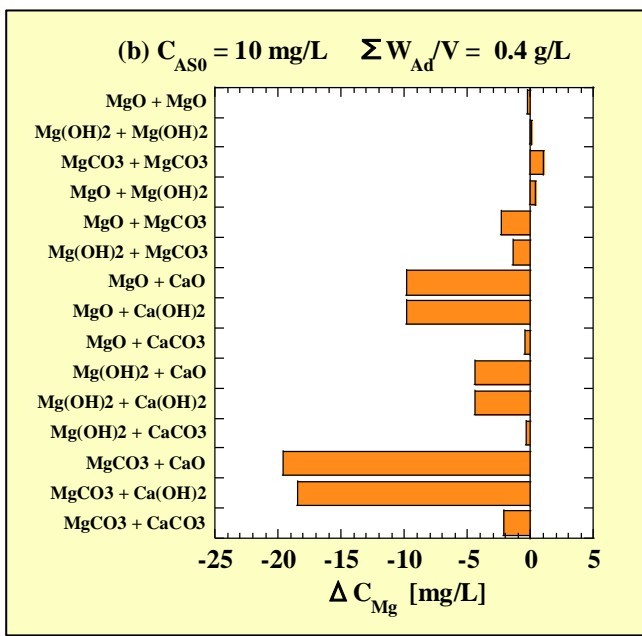

**Figure 4.** $\Delta C_{Mg}$ for combined-addition at (**a**) $C_{AS0}$ = 1 mg/L and (**b**) $C_{AS0}$ = 10 mg/L. $\Delta C_{Mg}$ = [the combined-addition $C_{Mg}$] − [the higher of the two single-addition $C_{Mg}$].

There was no significant difference in the tendency of $\Delta C_{Mg}$ depending on $C_{AS0}$. The $\Delta C_{Mg}$ of MgCO$_3$-CaO and -Ca(OH)$_2$ were the lowest followed by MgO-CaO and -Ca(OH)$_2$. For Mg(OH)$_2$-CaO and -Ca(OH)$_2$, the leaching of Mg was slightly inhibited. In contrast, the combined additions of the Mg-based adsorbents with CaCO$_3$ had almost no effect of inhibiting the leaching of Mg. In addition, no clear increase in the amount of Mg-leaching was observed even with the combinations between Mg-based adsorbents.

From the above, combining CaO or Ca(OH)$_2$ with the Mg-based adsorbents was shown to inhibit Mg-leaching. The effects of inhibiting Mg-leaching by CaO and Ca(OH)$_2$ were similar, and the magnitude of the effects was Mg(OH)$_2$ < MgO < MgCO$_3$.

*4.3. Ca-Leaching Behavior*

4.3.1. Ca-Residual Ratio

$R_{Ca}$ obtained from the As-removal tests in this study are shown in Tables 11–13. However, it should be noted that the values of $R_{Ca}$ obtained from these calculations include the resolidified portion of Ca once leached out, similar to $R_{Mg}$ for the Mg-based absorbent.

**Table 11.** Ca residual ratio ($R_{Ca}$) for single-addition As-removal tests.

| $W_{Ad}/V$ [g/L] | $C_{AS0}$ [mg/L] | MgO | Mg(OH)$_2$ | MgCO$_3$ | CaO | Ca(OH)$_2$ | CaCO$_3$ |
|---|---|---|---|---|---|---|---|
| 0.2 | 1 | - | - | - | 15.6 | 7.1 | 94.6 |
| 0.4 | 1 | - | - | - | 25.8 | 4.2 | 96.8 |
| 0.2 | 10 | - | - | - | 18.9 | 9.2 | 93.6 |
| 0.4 | 10 | - | - | - | 32.4 | 9.2 | 96.6 |

unit [%].

**Table 12.** Ca residual ratio ($R_{Ca}$) for combined-addition As-removal tests at $C_{AS0}$ = 1 mg/L.

| $\Sigma W_{Ad}/V$ = 0.4 g/L | MgO | Mg(OH)$_2$ | MgCO$_3$ | CaO | Ca(OH)$_2$ | CaCO$_3$ |
|---|---|---|---|---|---|---|
| CaO | 29.0 | 29.1 | 53.9 | 25.8 | 16.5 | 36.9 |
| Ca(OH)$_2$ | 9.7 | 10.2 | 72.5 | 16.5 | 4.2 | 47.2 |
| CaCO$_3$ | 96.6 | 97.0 | 98.6 | 36.9 | 47.2 | 96.8 |

unit [%].

**Table 13.** Ca residual ratio ($R_{Ca}$) for combined-addition As-removal tests at $C_{AS0}$ = 10 mg/L.

| $\Sigma W_{Ad}/V$ = 0.4 g/L | MgO | Mg(OH)$_2$ | MgCO$_3$ | CaO | Ca(OH)$_2$ | CaCO$_3$ |
|---|---|---|---|---|---|---|
| CaO | 31.3 | 35.6 | 42.3 | 32.4 | 25.4 | 44.4 |
| Ca(OH)$_2$ | 11.5 | 13.9 | 55.3 | 25.4 | 9.2 | 46.7 |
| CaCO$_3$ | 96.4 | 96.9 | 98.7 | 44.4 | 46.7 | 96.6 |

unit [%].

Table 11 shows $R_{Ca}$ for the single addition of one type of adsorbent obtained from As-removal tests.

Regardless of $C_{AS0}$, the $R_{Ca}$ of CaCO$_3$ exceeded 90%, but the $R_{Ca}$ of CaO and Ca(OH)$_2$ showed considerably low values. Even if the adsorbent addition concentration was doubled, the $R_{Ca}$ of Ca(OH)$_2$ and CaCO$_3$ hardly changed. In contrast, the $R_{Ca}$ of CaO nearly doubled when $W_{Ad}/$V was doubled.

Tables 12 and 13 show $R_{Ca}$ for the combined additions of two types of adsorbents obtained from As-removal tests for $C_{AS0}$ = 1 and 10 mg/L, respectively. For convenience, the results in the single additions of $W_{Ad}/V$ = 0.4 g/L have also been included in Tables 12 and 13, respectively. Similar to Figure 3, each $R_{Ca}$ of the combined additions is shown in Figure 5, based on the data in Tables 12 and 13. Figure 5a,b correspond to $C_{AS0}$ = 1 and 10 mg/L, respectively.

There was no significant difference in the tendency of $R_{Ca}$ depending on $C_{AS0}$. However, overall, $R_{Ca}$ tended to be slightly higher at $C_{AS0}$ = 10 mg/L than at $C_{AS0}$ = 1 mg/L. In combinations of CaCO$_3$ with the adsorbents other than CaO and Ca(OH)$_2$, the $R_{Ca}$ significantly exceeded 90%. The $R_{Ca}$ in most of the combinations of CaO and of Ca(OH)$_2$ were low overall; however, those with MgCO$_3$ were relatively high. The $R_{Ca}$ in the combinations of Ca(OH)$_2$, excluding those with MgCO$_3$, were considerably lower than those in the combinations of CaO and of CaCO$_3$.

### 4.3.2. Effects of Combined Addition on Ca-Leaching Behavior

Similar to Mg, even if the Ca-residual ratio is the same, the amount of Ca leached differs depending on the type of the Ca-based adsorbent used. Therefore, to evaluate the effects of the combined addition on the Ca-leaching behavior, it is considered more appropriate to compare the amount of change in $C_{Ca}$. Then, the higher $C_{Ca}$ in the single-addition test ($W_{Ad}/V$ = 0.2 g/L) of the two types of adsorbents used in the combined-addition test ($\Sigma W_{Ad}/V$ = 0.4 g/L) is subtracted from the $C_{Ca}$ in the combined-addition test. The obtained difference was denoted as $\Delta C_{Ca}$. The lower the value of $\Delta C_{Ca}$, the higher the effects of inhibiting the leaching of Ca.

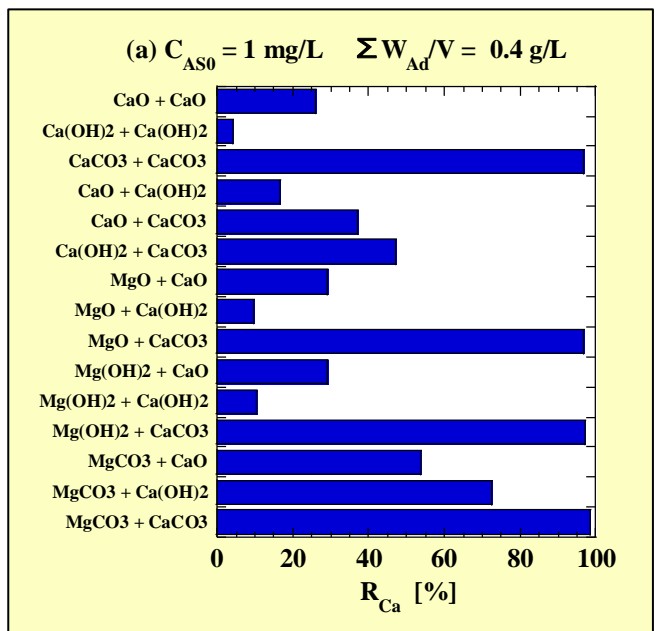
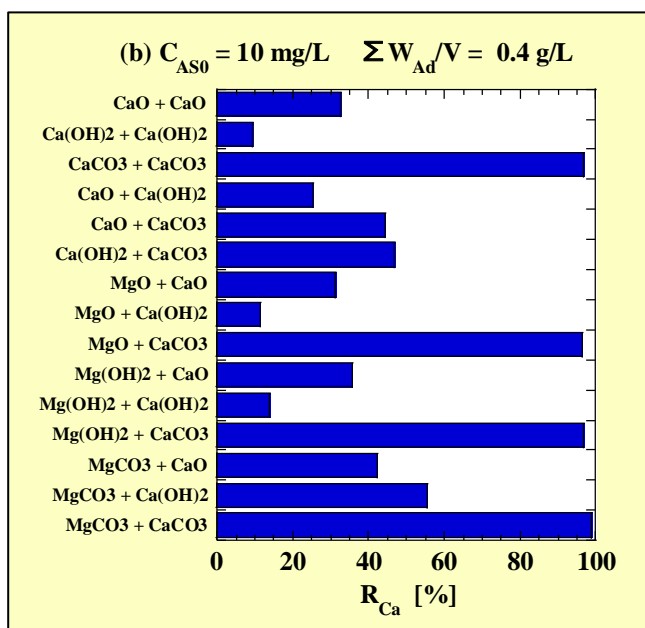

**Figure 5.** $R_{Ca}$ for combined-addition at (**a**) $C_{AS0}$ = 1 mg/L and (**b**) $C_{AS0}$ = 10 mg/L. $R_{Ca} = 100 - (C_{Ca} \times 1000)/(W_{Ad}/V \times (\alpha_{Ca}/100)) \times 100$.

The $\Delta C_{Ca}$ with the mass-based adsorbent addition concentration is shown in Table 14 for $C_{AS0}$ = 1 mg/L and 15 for $C_{AS0}$ = 10 mg/L, respectively. Similar to Figure 4, each $\Delta C_{Ca}$ of the combined additions is shown in Figure 6, based on the data in Tables 14 and 15. Figure 6a,b correspond to $C_{AS0}$ = 1 and 10 mg/L, respectively.

**Table 14.** The combined-addition $C_{Ca}$ minus the higher of the two single-addition $C_{Ca}$ ($\Delta C_{Ca}$) for As-removal tests at $C_{AS0}$ = 1 mg/L.

| $\Sigma W_{Ad}/V$ = 0.4 g/L | MgO | Mg(OH)$_2$ | MgCO$_3$ | CaO | Ca(OH)$_2$ | CaCO$_3$ |
|---|---|---|---|---|---|---|
| CaO | −24.1 | −24.2 | −62.9 | 84.3 | 88.6 | 16.8 |
| Ca(OH)$_2$ | 4.3 | −1.1 | −71.8 | 88.6 | 103 | −1.6 |
| CaCO$_3$ | 2.9 | 2.6 | 1.1 | 16.8 | −1.6 | 0.4 |

unit [mg/L].

**Table 15.** The combined-addition $C_{Ca}$ minus the higher of the two single-addition $C_{Ca}$ ($\Delta C_{Ca}$) for As-removal tests at $C_{AS0}$ = 10 mg/L.

| $\Sigma W_{Ad}/V$ = 0.4 g/L | MgO | Mg(OH)$_2$ | MgCO$_3$ | CaO | Ca(OH)$_2$ | CaCO$_3$ |
|---|---|---|---|---|---|---|
| CaO | −18.0 | −21.5 | −33.4 | 76.7 | 74.7 | 10.6 |
| Ca(OH)$_2$ | −2.9 | −4.7 | −50.0 | 74.7 | 102 | 2.9 |
| CaCO$_3$ | 2.9 | 2.4 | 1.1 | 10.6 | 2.9 | 0.2 |

unit [mg/L].

There was no significant difference in the tendency of $\Delta C_{Ca}$ depending on $C_{AS0}$. The $\Delta C_{Ca}$ of Ca(OH)$_2$-MgCO$_3$ was the lowest. The $\Delta C_{Ca}$ of CaO-MgCO$_3$, -Mg(OH)$_2$, and -MgO were the next lowest. In contrast, a clear increase in the amount of Ca-leaching was observed with the combinations between Ca-based adsorbents other than CaCO$_3$.

From the above, combining MgCO$_3$ with the Ca-based adsorbents was shown to inhibit Ca-leaching. It was not significantly different in the effects of inhibiting Ca-leaching by MgCO$_3$ between CaO and Ca(OH)$_2$. CaCO$_3$ also seemed to inhibit Ca-leaching by combining with MgCO$_3$; however, this is not clear because the $C_{Ca}$ values were very low.

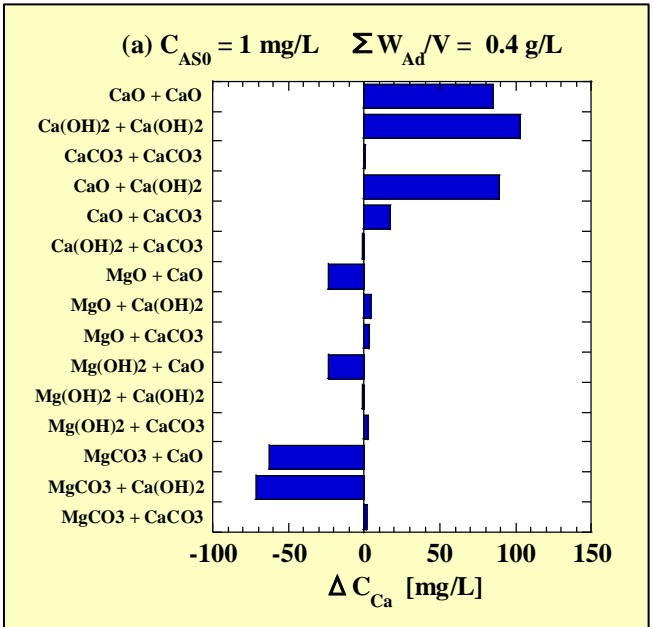
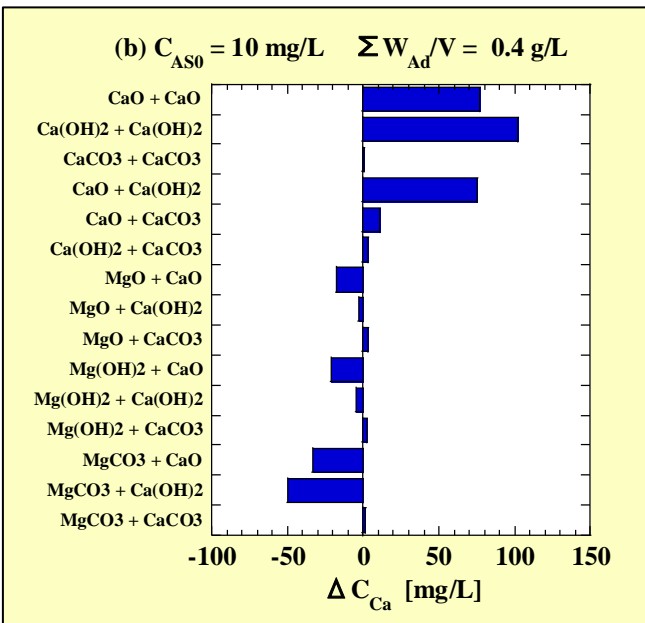

**Figure 6.** $\Delta C_{Ca}$ for combined-addition at (**a**) $C_{AS0}$ = 1 mg/L and (**b**) $C_{AS0}$ = 10 mg/L. $\Delta C_{Ca}$ = (the combined-addition $C_{Ca}$) − (the higher of the two single-addition $C_{Ca}$).

### 4.4. Mechanism of Inhibiting Mg- and Ca-Leaching

In this section, we have provided some brief considerations regarding the mechanisms of inhibiting Mg- and Ca-leaching by combining with Mg-based and Ca-based adsorbents.

First, the chemical reactions when each of the components are leached from the Mg-based and Ca-based adsorbents are shown below.

$$MgO\ (s) + H_2O\ (l) \longleftrightarrow Mg^{2+}\ (aq) + 2OH^-\ (aq) \tag{4}$$

$$Mg(OH)_2\ (s) \longleftrightarrow Mg^{2+}\ (aq) + 2OH^-\ (aq) \tag{5}$$

$$MgCO_3\ (s) \longleftrightarrow Mg^{2+}\ (aq) + CO_3^{2-}\ (aq) \tag{6}$$

$$CaO\ (s) + H_2O\ (l) \longleftrightarrow Ca^{2+}\ (aq) + 2OH^-\ (aq) \tag{7}$$

$$Ca(OH)_2\ (s) \longleftrightarrow Ca^{2+}\ (aq) + 2OH^-\ (aq) \tag{8}$$

$$CaCO_3\ (s) \longleftrightarrow Ca^{2+}\ (aq) + CO_3^{2-}\ (aq) \tag{9}$$

In reality, the carbonate ion in (6) and (9) undergoes the following chemical equilibrium reactions, which depend on pH.

$$CO_3^{2-}\ (aq) + H_2O\ (l) \longleftrightarrow HCO_3^-\ (aq) + OH^-\ (aq) \tag{10}$$

$$HCO_3^-\ (aq) + H_2O\ (l) \longleftrightarrow H_2CO_3\ (aq) + OH^-\ (aq) \tag{11}$$

The chemical reactions related to the inhibition of Mg- and Caleaching can be explained by the combination of the above chemical reactions.

#### 4.4.1. Chemical Reactions Involved in Inhibiting Mg-Leaching

In Section 4.2, the combined additions of the Mg-based adsorbents with CaO or Ca(OH)$_2$ were shown to inhibit Mg-leaching. $Mg^{2+}$ on each right side of Equations (4)–(6) reacts with $OH^-$ on each right side of Equations (7) and (8) to produce $Mg(OH)_2$.

$$Mg^{2+}\ (aq) + 2OH^-\ (aq) \longleftrightarrow Mg(OH)_2\ (s) \tag{12}$$

Therefore, one of the mechanisms for inhibiting Mg-leaching by the above-mentioned combined addition is the resolidification of the leached Mg owing to the production of $Mg(OH)_2$. In addition, the values of $pH_f$ were higher in the combined addition of CaO or $Ca(OH)_2$ than in the single additions of the Mg-based adsorbents; thus, the $Mg(OH)_2$ was more likely to be produced. In contrast, as the values of $pH_f$ in the combined addition with $CaCO_3$ added were almost the same as those in the single additions of the Mg-based adsorbents, indicating that $Mg(OH)_2$ could not be produced.

4.4.2. Chemical Reactions Involved in Inhibiting Ca-Leaching

In Section 4.3, the combined additions of the Ca-based adsorbents with $MgCO_3$ were shown to inhibit Ca-leaching. $Ca^{2+}$ on each right side of Equations (7)–(9) reacts with $CO_3^{2-}$ on each right side of Equation (6) to produce $CaCO_3$.

$$Ca^{2+} \text{ (aq)} + CO_3^{2-} \text{ (aq)} \longleftrightarrow CaCO_3 \text{ (s)} \tag{13}$$

Therefore, one of the mechanisms for inhibiting Ca-leaching by the above-mentioned combined addition is the resolidification of the leached Ca due to production of $CaCO_3$. $Ca(OH)_2$ may also be considered a candidate for resolidified substances. However, as the $pH_f$ is lower in the combined addition with $MgCO_3$ than in the single additions of CaO or $Ca(OH)_2$, it is considered unlikely that $Ca(OH)_2$ was produced.

In addition, compounds such as $(Mg)_p(Ca)_q(OH)_r(CO_3)_s$ (p, q, r and s are positive numbers) may be produced; it is thus necessary to perform highly accurate X-ray analysis and detailed chemical equilibrium calculation in the future.

*4.5. Mechanism of As Adsorption*

4.5.1. Dissolved Forms of Arsenic in Solution

The dissolved forms of As(V) in solutions are represented by the following dissociation reaction formulas for arsenic acid:

$$H_3AsO_4 \text{ (aq)} \longleftrightarrow H_2AsO_4^- \text{ (aq)} + H^+ \text{ (aq)} \tag{14}$$

$$H_2AsO_4^- \text{ (aq)} \longleftrightarrow HAsO_4^{2-} \text{ (aq)} + H^+ \text{ (aq)} \tag{15}$$

$$HAsO_4^{2-} \text{ (aq)} \longleftrightarrow AsO_4^{3-} \text{ (aq)} + H^+ \text{ (aq)} \tag{16}$$

where $pKa_1 = 2.24$, $pKa_2 = 6.96$, and $pKa_3 = 11.50$ (25 °C) [39].

The value of $pH_0$ was 6.91–7.32 (see Tables S2 and S4–S6) and $pKa_1 \ll pH_0 \ll pKa_3$, so the abundances of $H_3AsO_4$ and $AsO_4^{3-}$ were considered to be extremely small and negligible. The value of $pKa_2$ was applied to Equation (15). Then, after substituting the $pH_0$ value, the value of $(HAsO_4^{2-})/(H_2AsO_4^-) = 10 \exp (pH_0 - pKa_2)$ was calculated to be 0.89–2.29. Based on the results, the main dissolved forms of As in the simulated As(V)-contaminated water before adding the adsorbents were presumed to be $H_2AsO_4^-$ and $HAsO_4^{2-}$.

On the other hand, the value of $pH_f$ was 9.45 to 12.05, and differed greatly depending on the type and combination of adsorbents (see Tables S2 and S4–S6). $pKa_2 \ll pH_f$, so the abundances of $H_3AsO_4$ and $H_2AsO_4^-$ were considered to be extremely small and negligible. In addition, in the cases of the single addition, the values of $pH_f$ for $CaCO_3$, $Mg(OH)_2$, $MgCO_3$, MgO, $Ca(OH)_2$, and CaO were 9.45–9.71, 10.39–10.48, 10.59–10.68, 10.81–10.94, 11.72–12.05, and 11.80–12.05, thus the values of $(AsO_4^{3-})/(HAsO_4^{2-}) = 10 \exp (pH_f - pKa_3)$ were $\ll 0.1$, $< 0.1$, 0.12–0.15, 0.20–0.26, 1.66–3.55, and 2.00–3.55, respectively. Therefore, the main dissolved forms of As in the treated water were inferred to be $HAsO_4^{2-}$ for MgO, $Mg(OH)_2$, $MgCO_3$, and $CaCO_3$, and then $AsO_4^{3-}$ for CaO and $Ca(OH)_2$.

However, as the dissolved form of As would change accordingly as the pH of the solution increases with the leaching (dissolution) of the base material components from the adsorbents, any form of As that may be present at the pH range of $pH_0$ to $pH_f$ could react with the adsorbent. In addition, even in the case of the combined addition, the

idea regarding the dissolved form of As is the same as in the case of the single addition described above.

### 4.5.2. Reactions of Adsorption, Precipitation, and Co-Precipitation of As

It is considered that a part of each adsorbent is dissolved in the solution by the reaction of Equations (4)–(9) shown in Section 4.4, and the adsorbent surface remaining as a solid phase is also hydrated to form Solid-Me-OH, where Me is Mg or Ca.

First, assuming $H_2AsO_4^-$ as the adsorbed species, the reaction with the adsorbent is speculated below.

$$\text{Solid-Me-OH (s)} + H_2AsO_4^- \text{ (aq)} \longleftrightarrow \text{Solid-Me-O-AsO(OH)}_2 \text{ (s)} + OH^- \text{ (aq)} \qquad (17)$$

Equation (17) represents the chemical adsorption reaction due to the ion exchange between $H_2AsO_4^-$ and $OH^-$.

Additionally, the adsorption reactions after the hydroxyl groups on the surface of the adsorbent dissociate to form Solid-Me$^+$ are considered:

$$\text{Solid-Me}^+ \text{ (s)} + H_2AsO_4^- \text{ (aq)} \longleftrightarrow \text{Solid-Me-O-AsO(OH)}_2 \text{ (s)} \qquad (18)$$

Equation (18) represents the electrostatic chemical adsorption reaction.

In both Equations (17) and (18), $H_2AsO_4^-$ on the left sides can be replaced with $HAsO_4^{2-}$ or $AsO_4^{3-}$. In those cases, Solid-Me-O-AsO(OH)$_2$ on the right sides in Equations (17) and (18), are replaced by Solid-Me-O-AsO(OH)O$^-$ and Solid-Me-O-AsO(O$^-$)$_2$. However, it is difficult to identify the actual adsorbed arsenic acid species as mentioned in Section 4.5.1 and also to differentiate between Equations (17) and (18).

Furthermore, a reaction in which released $Mg^{2+}$ or $Ca^{2+}$, as in Equations (4)–(9), binds with arsenic acid species to form and precipitate arsenate species is also conceivable. For example, assuming $AsO_4^{3-}$ as the reacting species, the precipitation reaction of arsenate species is described as:

$$3Me^{2+} \text{ (aq)} + 2AsO_4^{3-} \text{ (aq)} \longleftrightarrow Me_3(AsO_4)_2 \text{ (s)} \qquad (19)$$

However, the formation of arsenate species also significantly depends on the concentration of each dissolved component and pH; thus, the possibility of the formation should be verified by chemical equilibrium calculations.

$R_{AS}$ for the combined addition of each MgCO$_3$-CaO and MgCO$_3$-Ca(OH)$_2$ was much higher than that for the single addition of each MgCO$_3$, CaO and Ca(OH)$_2$ as described in Section 4.1. This is due to As adsorption on MgCO$_3$, CaO, and Ca(OH)$_2$ individually but also to As being incorporated and co-precipitated when Mg(OH)$_2$ and CaCO$_3$ are produced, as in Equations (12) and (13).

### 4.6. Comprehensive Evaluation

To summarize what has been described above, as evaluated in terms of both As-removal performance and leaching of the base material components from the adsorbent, the single addition or combined addition of the adsorbents, which would be highly evaluated comprehensively, are as follows.

At $C_{AS0}$ = 1 mg/L, the most performant strategies are the single addition of MgO or Mg(OH)$_2$, or the combined additions of Mg(OH)$_2$-CaO, MgCO$_3$-CaO or MgCO$_3$-Ca(OH)$_2$. At $C_{AS0}$ = 10 mg/L, the most performant strategies are MgCO$_3$-Ca(OH)$_2$ or MgCO$_3$-CaO. Therefore, if the As(V)-contaminated water may have a wide concentration range, MgCO$_3$-Ca(OH)$_2$ or MgCO$_3$-CaO are the recommended adsorbent compositions.

Since Fe-based adsorbents are expensive, regeneration treatment with Ca-based adsorbents has been suggested [34], which can lead to increased operational complexity and cost. In addition, it has been pointed out that when Fe(II) exists on the surface of Fe-based adsorbents, the adsorbed As(V) may be reduced to highly toxic As(III) and eluted [25].

Such a reduction reaction would not occur for Mg-based and Ca-based adsorbents. Furthermore, adsorbents doped with cobalt (one of transition elements) have problems with cobalt leaching [29], which may lead to health risks. On the other hand, Mg and Ca components that may leach from Mg-based and Ca-based adsorbents are not harmful to humans or animals. Since it is necessary to select the optimum adsorbent according to the contamination state, contamination factors, purification conditions, etc., it would be better to list a wide variety of adsorbents as candidates in advance. This study reported the results of arsenic-removal tests with various combinations of Mg-based and Ca-based adsorbents, and it is hoped that the results of this research will be useful when selecting a combination of adsorbents according to the arsenic contamination state.

In this study, the As-removal tests used two types of adsorbents combined at an adsorbent addition concentration of 1:1 by mass ratio. In the near future, stoichiometric and chemical equilibrium studies should be conducted and the effect of inhibiting the leaching should be analyzed using the data obtained in this study. Additional tests on the molar ratio of the adsorbent addition concentration as variables are needed, and an optimum adsorbent mixing ratio should be derived. Furthermore, in future research, we also plan to investigate this process using "arsenite" rather than "arsenate".

## 5. Conclusions

In this study, As-removal tests with the combined addition of Mg- and Ca-based adsorbents were systematically conducted for the purpose of improving the As-removal performance and inhibiting the leaching of base material components from adsorbents (i.e., improving the environmental stability of adsorbents). The results of the combined addition tests were compared to those of the single-addition tests with each adsorbent.

Many of the combined additions tested in this study, especially $MgCO_3$-CaO and $MgCO_3$-$Ca(OH)_2$, are promising. Some combined additions of the Ca-based adsorbents, MgO and -$Mg(OH)_2$, decreased the As-removal performance. The As-removal performance of the combined addition of $MgCO_3$-$Ca(OH)_2$ was the highest at both 1 and 10 mg/L of $C_{AS0}$. It was clarified that the combined additions of the Mg-based adsorbents with CaO or $Ca(OH)_2$ inhibited Mg-leaching. The effects of inhibiting Mg-leaching by CaO and $Ca(OH)_2$ were similar but depended on the Mg species, with $Mg(OH)_2$ < MgO < $MgCO_3$. One of the mechanisms for inhibiting Mg-leaching via these combined additions was considered to be the resolidification of the leached Mg owing to the production of $Mg(OH)_2$. It was also clarified that the combined addition of CaO or $Ca(OH)_2$ with $MgCO_3$ inhibited the leaching of Ca. It was not significantly different in the effects of inhibiting the Ca-leaching by $MgCO_3$ between CaO and $Ca(OH)_2$. One of the mechanisms for inhibiting Ca-leaching by these combined additions was the resolidification of the leached Ca owing to the production of $CaCO_3$. The improvement of the As-removal ratio for the combined addition of each $MgCO_3$-CaO and $MgCO_3$-$Ca(OH)_2$ was considered to be caused by the incorporation and co-precipitation with As when $Mg(OH)_2$ and $CaCO_3$ were produced, in addition to As adsorption on each adsorbent. From the viewpoints of both As-removal performance and the environmental stability of the adsorbents, $MgCO_3$-$Ca(OH)_2$ or $MgCO_3$-CaO are recommended as improved sustainable adsorbent combinations that can be effectively applied over a wide range of As concentrations.

Since the As-removal tests in this study were performed at only two initial As-concentrations and one initial pH, we were not possible to determine the optimal range for As-removal performance. Although the combined-addition tests in this study were performed at only an adsorbent addition concentration of 1:1 by mass ratio, in order to derive the optimum mixing ratio of the adsorbents, it will be necessary to conduct additional tests with varying the adsorbent mixing ratio as a parameter. Nevertheless, this study demonstrated that the combined addition of specific Mg-based and Ca-based adsorbents is possible to improve the As-removal performance and also inhibit the leaching of the matrix component from the adsorbents. The results of this study are expected to increase the options for using Mg-based and Ca-based adsorbents for arsenic contamination treatment.

**Supplementary Materials:** The following supporting information can be downloaded at: https://www.mdpi.com/article/10.3390/su15054689/s1, Table S1: characteristics of the adsorbents where $\alpha$ is the measured content, P is the reagent purity, Dp50 is the median particle size, and SBET is the Brunauer–Emmett–Teller (BET) surface area; Table S2: target and measured values for the $C_{AS0}$ = 1 mg/L, single-addition tests where one type of adsorbent was employed at a time to assess As removal; Table S3: relative standard errors for test nos. 1–6 in Table S2; Table S4: target and measured values for the $C_{AS0}$ = 10 mg/L, single-addition tests where one type of adsorbent was employed at a time to assess As removal; Table S5: target and measured values for the $C_{AS0}$ = 1 mg/L, combined-addition tests where two types of adsorbents were added at a time to assess As removal; Table S6: target and measured values for the $C_{AS0}$ = 10 mg/L, combined-addition tests where two types of adsorbents were added at a time to assess As removal.

**Author Contributions:** Conceptualization, H.S.; methodology, H.S. and J.H.; formal analysis, H.S. and J.H.; investigation, T.O. and H.S.; resources, H.S., M.Z., and J.H.; data curation, T.O. and H.S.; writing—original draft preparation, H.S. and T.O.; writing—review and editing, H.S., T.O., M.Z., J.H. and Y.K.; supervision, H.S. and M.Z.; project administration, M.Z. and Y.K.; funding acquisition, M.Z., Y.K., and H.S. All authors have read and agreed to the published version of the manuscript.

**Funding:** This research received no external funding.

**Institutional Review Board Statement:** Not applicable.

**Informed Consent Statement:** Not applicable.

**Data Availability Statement:** Data that support the findings of this study are available from the corresponding authors upon reasonable request.

**Acknowledgments:** We are deeply grateful to Norihisa Fukaya (Interdisciplinary Research Center for Catalytic Chemistry, AIST) for measuring the BET specific surface area of $CaCO_3$.

**Conflicts of Interest:** The authors declare no conflict of interest.

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
