# Peer review of "Removal of Arsenate from Contaminated Water via Combined Addition of Magnesium-Based and Calcium-Based Adsorbents"

_sustainability, doi:10.3390/su15054689_

Round 1

Reviewer 1 Report

Overall, the paper is well written, especially the way of presenting data is impressive. However, there are a few things that still need to be addressed. Therefore, I recommend that minor revision is warranted.

Detailed comments are listed below.

(1) Page 3, Lines 110-111. Grammatical error. I noticed some other grammatical errors as well. Please check carefully throughout the manuscript and correct them all.

(2) Page 3, Lines 115-116. Please briefly introduce these studies, for instance, the key findings. You may also cite them as you discuss the single addition of Mg- and Ca-based adsorbents in Section 4.

(3) Page 4, Line 148. What is the criterium of selecting these two initial As concentrations? What is their environmental relevance, i.e., what is the typical range of As levels in contaminated water?

(4) Page 4, Lines 172-179. Consider combining this section with Section 2.3.

(5) Page 4, Lines 176-179. Again, what is the criterium of selecting the dosage of the adsorbents used in this study? Is 0.2 or 0.4 g/L within the typical range in real-world applications?

(6) Page 13, Line 477. A general question: any comments on the adsorption isotherm models that the experimental results from this work would align with?

(7) Page 15, Line 548. This study investigated single addition and combined addition of two types of adsorbents. Could you provide a prediction for situations with a mixed addition of MORE THAN TWO types of adsorbents?

Reviewer 2 Report

The manuscript, entitled “Removal of Arsenate from Contaminated Water via Combined Addition of Magnesium-Based and Calcium-Based Adsorbents”, investigated the effects of the combined addition of Mg- and Ca-based adsorbents arsenic removal performance and inhibiting leaching of base material components from the adsorbent. This study is innovative and the data are well collected. I would like to ask the authors to revise the following points before accepting it for publication.

1.      Figures are missing in the whole manuscript. Appropriate use of figures can make the article more attractive. Graphical abstract and other figures are suggested in this manuscript.

2.      The introduction part is poorly designed and written. Many other As treatment methods except adsorption are not necessarily described in detail. For example, lines 50-62, and 77-105 are not necessary. It is suggested to rewrite the introduction part.

3.      Related papers have been reported by different research groups. It is better to cite the following references to support some related paragraphs in the introduction.

Environmental Science & Technology, 2015, 49(13): 7726-7734.

Journal of Environmental Sciences, 2022, 121: 1-12.

Journal of Environmental Sciences, 2022, 122: 217-226.

4.      In Table 2, the RAS of MgO are higher than the RAS of Mg(OH)2 in all parameters except when WAd/V = 0.2, CAS0 = 1; also, in the case of the CaO and Ca(OH)2. Could the authors try to explain the reasons for the different tendency?

5.      The authors give the arsenic removal ratio in each experiment. However, the adsorption capacities under optimized conditions are suggested to be compared with other related works, especially the widely reported iron-based adsorbents.

6.      Surface area is an important parameter of adsorbent in the adsorption process. In the discussion part, its effect should be considered.

7.      The manuscript should be simplified. For example, Table 1 could be moved to the SI part. The introduction of Mg-Residual Ratio (lines 314-317) and Ca-Residual Ratio (lines 376-379) should be combined and moved to the Materials and Methods part. 

Reviewer 3 Report

The results of this research are conveyed thoughtfully and completely, and they are consistent with the experimental findings. However, the authors failed to explain and draw out the novelty of the work, this aspect needs to be improved. This work is worthwhile to be publish in this journal after major revision. The following issues should be addressed:

1. Maybe the author should compare their results clearly with other reported works, highlighting the advantage and disadvantages of their novel materials.

2. Introduction is not well-organized and the importance and novelty of the research should be highlighted and more clearly stated. The authors should give some examples of works in the bibliography, to clear the advantage of their work in comparison with those works.

3. The authors are responsible for the English, which should be polished throughout the manuscript to clear some minor typo/grammar errors.

4. Introduction part, if possible, some important and relative reports references could helped: https://doi.org/10.1016/j.surfin.2022.102006, https://doi.org/10.1016/j.colsurfa.2021.127753, https://doi.org/10.1016/j.desal.2023.116377.

5. Abstract and conclusion not targeted; the authors should rephrase it.

6. In Materials and method section, please provide the purity of your chosen precursors.

7. The author should better improve the beauty and quality of the tables.

8. In experimental part, the authors should provide a reference on how they prepare the adsorbents.

Round 2

Reviewer 3 Report

Accepted in present form

Author Response

We are very grateful for your acceptance of our revised manuscript.

Your suggestions and comments have greatly improved our manuscript.

Thank you so much.

Sincerely,

Hajime Sugita